# Development on Infected Citrus over Generations Increases Vector Infection by ‘*Candidatus* Liberibacter Asiaticus in *Diaphorina citri*’

**DOI:** 10.3390/insects11080469

**Published:** 2020-07-24

**Authors:** Inaiara de Souza Pacheco, Diogo Manzano Galdeano, João Roberto Spotti Lopes, Marcos Antonio Machado

**Affiliations:** 1Centro de Citricultura Sylvio Moreira, Instituto Agronômico de Campinas, Cordeirópolis, São Paulo 13490-970, Brazil; diogo_manzano@hotmail.com (D.M.G.); marcos@ccsm.br (M.A.M.); 2Instituto de Biologia, Universidade Estadual de Campinas, Campinas, São Paulo 13083-862, Brazil; 3Departamento de Entomologia e Acarologia, Escola Superior de Agricultura “Luiz de Queiroz”—Universidade de São Paulo, Piracicaba, São Paulo 13418-900, Brazil; jrslopes@usp.br

**Keywords:** citrus Huanglongbing, vector-borne bacterium, Asian citrus psyllid, vector infectivity

## Abstract

‘*Candidatus* Liberibacter asiaticus’ (CLas) is a major causal agent of citrus Huanglongbing (HLB), which is transmitted by Asian citrus psyllid (ACP), *Diaphorina citri*, causing severe losses in various regions of the world. Vector efficiency is higher when acquisition occurs by ACP immature stages and over longer feeding periods. In this context, our goal was to evaluate the progression of CLas population and infection rate over four ACP generations that continuously developed on infected citrus plants. We showed that the frequency of CLas-positive adult samples increased from 42% in the parental generation to 100% in the fourth generation developing on CLas-infected citrus. The bacterial population in the vector also increased over generations. This information reinforces the importance of HLB management strategies, such as vector control and eradication of diseased citrus trees, to avoid the development of CLas-infected ACP generations with higher bacterial loads and, likely, a higher probability of spreading the pathogen in citrus orchards.

## 1. Introduction

Huanglongbing disease is the most destructive bacteriosis in the citrus industry around the world. This disease is characterized by yellow and mottled shoots, with leaves presenting chlorotic spots similar to nutritional deficiencies. Huanglongbing (HLB)-affected fruits present color inversion, distortion of the central columella and aborted seeds. The progression of symptoms culminates in intense defoliation and fast tree decline [1].

Asian citrus psyllid (ACP), *Diaphorina citri* Kuwayama (Hemiptera: Liviidae), is the natural vector of ‘*Candidatus* Liberibacter asiaticus’ (CLas), the main bacterium associated with HLB across citrus crops of the Americas and Asia. ACP transmits CLas in a circulative-propagative manner, which involves pathogen acquisition during phloem-sap ingestion, movement and multiplication in vector organs, followed by inoculation via saliva [2,3,4]. Both ACP nymphs and adults can acquire CLas, but this pathogen multiplies faster, reaches higher populations and is more efficiently transmitted in psyllids that acquire it during immature stages [5,6].

CLas is systemic, but is irregularly distributed in both ACP and citrus tissues [3,7]. The fastidious development of CLas results in delayed emergence of symptoms, making early HLB detection very difficult [1,8]. The long incubation period in citrus favors HLB spread, since the maintenance of infected plants in citrus orchards constitutes a source of CLas inoculum [8,9]. ACP can acquire and transmit the pathogen from newly-infected citrus trees before symptoms appear [10].

Tri-trophic interactions among pathogens, vectors and hosts highlight the complexity of vector-borne diseases such as HLB. The ability of pathogens to establish long-lasting associations with vectors may result in pathogen reservoirs within its vectors, independent from host plants, with significant impacts on pathogen dissemination [11]. Diverse parameters of pathogen–vector interaction, such as acquisition, latency periods and pathogen propagation, can interfere with pathogen transmission and may influence disease dynamics. Quantifying transmission parameters is essential for determining vector efficiency in spreading the pathogen and for establishing reliable epidemiological models [11,12,13].

The transmission parameters of CLas by ACP are a controversial topic of the HLB pathosystem. Efficient acquisition time ranges from hours to weeks [2,5]. Latency and inoculation periods have not been well established [2,6,14]. On the other hand, it has been hypothesized that longer acquisition access periods may result in a higher frequency of CLas-positive psyllids with high bacterial loads [5,15]. Based on this information, this work aimed to determine the progression of CLas infection levels across several *D. citri* generations.

## 2. Materials and Methods

### 2.1. Biological Material

The ACP used in this experiment came from a CLas-free ACP colony established on orange jasmine (*Murraya paniculata* (L.) Jack, Rutaceae) plants inside screened cages in a climate-controlled room at 25 ± 2 °C, under a 14:10 h (light:dark) photoperiod and 60–70% relative humidity. Psyllid samples were periodically tested by qPCR to confirm that the colony remained free of CLas.

Six-month-old sweet orange trees (*C. sinensis* cv. Pera on *C. limonia* Osbeck rootstock) were CLas graft-inoculated and kept in a greenhouse for two years. After this period, these plants (HLB-symptomatic and PCR positive—Ct values ranging from 30 to 32) were used as CLas source plants in the following experiment.

### 2.2. CLas Acquisition Bioassay over Multiple ACP Generations

Five CLas-infected and symptomatic sweet orange plants were kept separately inside screened cages, each plant constituting one biological replicate. In each cage, groups of 14 CLas-free ACP adults within 10 days of emergence were maintained for an oviposition period (OP) of 15 days (Figure 1A). After this period, parental ACPs were removed from cages and stored at −80 °C (Figure 1B). The progeny development was monitored daily and the emerging adults were kept on CLas-infected plants until the observation of the next egg generation (20 days post-emergence—PE) (Figure 1B,C), when all adults were removed (and stored at −80 °C) and the progeny development was monitored (Figure 1C,D). This process was repeated four times (four ACP generations). This acquisition bioassay was conducted under greenhouse conditions of light (10–13 h of light), with temperatures ranging from 17 to 35 °C and a relative humidity of 40–90%.

### 2.3. Nucleic Acid Extraction and CLas Detection by qPCR in Psyllid Samples

After CLas acquisition bioassay over multiple ACP generations, samples composed of a pool of five ACP adults were used for DNA extraction as described by Coletta-Filho et al. (2014). Integrity and quality of extracted nucleic acids were verified using NanoDrop ND 8000 spectrophotometer (NanoDrop Technologies, Wilmington, DE, USA) and 1% agarose gel electrophoresis.

The detection of CLas was performed using 120 nM of RpoB primers, Sybr qPCR MasterMix™ (Promega, Madison, WI, USA), and 100 ng of total DNA. Amplification cycles were performed on 7500 Fast Real-Time PCR System device (Thermo Scientific, Waltham, MA, USA), using the standard thermal profile: 95 °C for 20 s, followed by 40 cycles of 95 °C for 30 s, and 60 °C for 30 s. Three technical replicates were analyzed for each sample. Fifteen to 34 samples per ACP generation were analyzed.

The RpoB primers targeting DNA-directed RNA polymerase subunit beta gene (AWL13590.1) of CLas: RpoB-F 5′ CGTGGGCCTGGAGTTGT 3′ and RpoB-R 5′ GCGCACTTTCCACCGTAT 3′.

### 2.4. Molecular Standard Curve

In order to produce standard positive controls, conventional PCR was performed using 120 nM of RpoB primer pair, 25 µL of GoTaq Colorless Master mix (Promega) and 100 ng of total DNA (50 µL final reaction volume). The thermal profile used was: 94° for 3 min followed by 35 cycles of 94 °C for 30 s, 60 °C for 30 s, 72 °C for 1 min, and a final extension of 72 °C for 3 min. Amplicons were observed on 1% agarose electrophoresis gel and the product was purified using Wizard^®^ SV Gel and PCR Clean-Up System (Promega, Madison, WI, USA) according to the manufacturer’s instructions. This fragment was ligated to pGen-T Easy vector (3015 pb) (Promega) and transformed into DH5-α *Escherichia coli* cells according to the manufacturer’s instructions. Plasmid DNAs were extracted from transformed cells using PureYield™ Plasmid Miniprep System (Promega, Madison, WI, USA).

A standard curve was obtained by using 2.8 × 10^9^ copies from the purified plasmid, which were sequentially diluted at 10-fold intervals. A second curve was generated using 100 ng of genomic DNA from healthy ACP, mixed with the plasmid dilution to simulate natural conditions.

The proportion of CLas positive insects per generation was estimated by the formula *p* = 1 − I^1/k^, where I is the frequency of non-infected insects and k is the number of insects used per sample [16]. The bacterial load across *D. citri* generations was evaluated by one-way ANOVA (Tukey test) statistical analysis performed using GraphPad Prism software [17]. A normality test confirmed normal data distribution, which presented no violations of the ANOVA assumptions.

## 3. Results and Discussion

The standard curve generated for CLas-RpoB gene showed a linear relationship between the log of DNA copies and qPCR cycles (Figure 2), presenting an amplification efficiency of 1 based on the equation AE = (10^−1/slope^) − 1. No significant difference was observed between curves with and without ACP DNA (data not shown), indicating that the extracted genomic DNA did not present an inhibitory effect on the detection. Based on this curve, samples with Ct ≥ 34 were considered CLas-negative.

Under continuous ACP reproduction and development on CLas*-*source plants, it was observed that the frequency of qPCR-positive samples for this bacterium increased across psyllid generations. In addition to the increased frequency of CLas*-*positive samples, significant elevation in the bacterial load of infected adults was also observed (Table 1).

After 15 days of exposure to CLas-infected plants, the frequency of CLas-positive parental ACPs was 40% (Table 1). The low frequency of CLas-infected samples of parental ACP adults is consistent with previous reports of 40–50% infection rates in psyllids submitted to an acquisition access period of 15 days on source plants of this pathogen [5,18,19]. Moreover, it was observed that CLas-positive adults of the parental generation showed a low pathogen load (4.79 × 10^0^ bacterial cells/100 ng of DNA) (Table 1).

Less efficient CLas acquisition by ACP adults in comparison with nymphs has been reported [5,19]. ACP adults have fully developed immune systems, which may activate a strong defense response in comparison with immature stages [20,21]. Suppression of metabolic genes and apoptosis induction were observed in CLas-infected adults of *D. citri*, which could indicate an effort to limit bacterial spread [22,23,24]. In this context, the more efficient defense response to CLas infection observed in ACP adults may result in less efficient acquisition rates in the adult stage, which may explain the lower frequency of CLas-positive samples of parental generation psyllids in the present study.

The first ACP generation (F1) that developed from eggs to adults on CLas-infected plants showed an increase in the frequency of CLas-positive insect samples (66%) and in the bacterial load (1.40 × 10^1^/100 ng of DNA) (Table 1). Likewise, it has been reported that *D. citri* adults which emerged from CLas-infected nymphs showed higher CLas load [5,18,19]. Efficient acquisition by nymphal instars may be a result of many factors. For example, *D. citri* nymphs spend more time feeding on plants compared to ACP adults, which corroborates increased acquisition rates [25]. A further increase in the frequency of CLas-positive samples (86%) was observed in the second fully developed ACP generation (F2) on infected citrus and also an increase in the bacterial load in comparison with the F1 generation (6.85 × 10^1^/100 ng of DNA) (Table 1). In the fourth generation (F4), 100% of ACP samples were CLas-infected, showing an even higher bacterial load (5.39 × 10^3^/100 ng of DNA, Table 1). On the other hand, by applying the Swallow methodology (1987), the proportion of CLas acquisition to single insects (*p*) in each generation were low, as shown in Table 1. The proportion of positive insects was 9%, 24% and 35% for parental, F1 and F2 generations, respectively. Once 100% of analyzed samples were CLas-positive in the F4 generation, the *p* parameter was not calculated in order to avoid biased estimation—see Table 1.

This increase in the CLas population across ACP generations may be explained by the efficient acquisition by nymphal instars, which are less mobile than ACP adults and spend more time feeding on plants [25]. In addition, gene expression and microscopy studies indicate that CLas colonizes ACP nymphs more efficiently than adults [26,27,28].

The increment in bacterial load across ACP generations fully developed on CLas-infected citrus trees could also be explained by the circulative-propagative nature of CLas in the psyllid vector. After the acquisition from an infected plant, CLas spread across ACP organs and tissues, remaining within the insect for long periods of time, even after molting from one nymphal instar to the next and from nymphs to adults. Furthermore, sexual and transovarial CLas transmission has been reported, which may also contribute to increase in bacterial levels across ACP generations [3,19,29,30]. Studies have shown that CLas load in the vector rises faster when ACP acquires CLas as nymphs rather than as adults, indicating that CLas multiplies more efficiently in immature stages than in adults [5,6,18,31]. Longer acquisition access periods also increase the bacterial load in the vector, as shown for CLas in ACP (particularly for acquisition by adults) [5], and for *Candidatus* Liberibacter solanacearum (*CLso*) in t potato psyllid, *Bactericera cockerelli* (Sulc) [32].

Similarly to the results observed here, a preliminary study reported by Ammar and collaborators showed CLas-positive ACP frequency ranging from 93% to 97% when insects are reared on CLas-infected citron plants for several generations [33].

The same report showed that, in comparison with insects continuously maintained on CLas-positive citrons, ACPs maintained on *Murraya* plants for six months and then transferred to CLas-positive citrons for two generations presented low rates of CLas-positive insects. These results indicate that *Murraya*-reared ACPs may perform an antimicrobial activity compared to ACPs reared on citrus plants [33]. On the other hand, studies have shown that different citrus genotypes can interfere with ACP development, but not the CLas acquisition process [31,34].

Despite this, the host-switching, long-term relationship between the ACP population and CLas-infected plants, could result in a more stable association between these organisms [35]. ACP nymphs present an attenuated bacterial response that allows efficient CLas colonization [27,36]. Assuming that immune suppression is an evolutionary response to maintain chronic infections with endosymbiont organisms in hemipterans [37,38], it could be hypothesized that this long-term interaction could reinforce the ties between CLas and ACP. However, more investigations are necessary to confirm these findings.

Likewise, the transmission efficiency of CLas is influenced by *D. citri* age and duration of acquisition access periods in infected plants [39]. ACPs that acquire CLas during nymphal stages can transmit this bacterium more successfully than those whose acquisition occurred in the adult phase [6,9]. Although CLas transmission rates were not evaluated in the present study, it is possible that rearing *D. citri* on CLas-infected plants over several generations could result in insects being able to transmit CLas more efficiently.

Natural variation on CLas infectivity among ACP populations has been observed in some studies [2,39] and the maintenance of ACP colonies under laboratory conditions for long periods could interfere on the acquisition and inoculation of CLas by ACP [2]. Thus, further studies should be carried out verify how different ACP populations respond to long-term interactions with CLas-infected plants.

## 4. Conclusions

In conclusion, it was observed that the proportion of CLas-infected ACPs and the bacterial load both increase when ACPs are reared on CLas-infected plants over several generations. The maintenance of ACPs on HLB-positive plants for long periods also results in elevated CLas load in the vector, possibly increasing transmission efficiency and the rates of disease dissemination in citrus orchards. This information reinforces the importance of HLB management strategies focused on the eradication of CLas-infected plants and control of vector populations developing on infected plants within orchards or externally.

## Figures and Tables

**Figure 1 insects-11-00469-f001:**
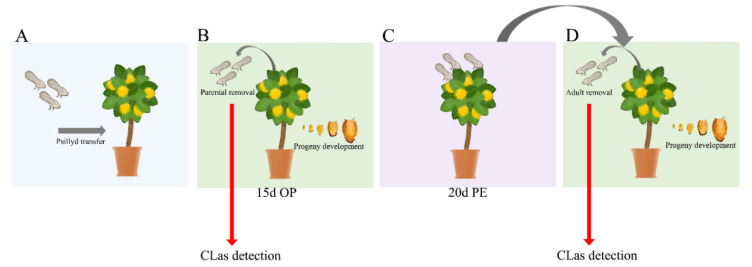
Scheme of *Candidatus* Liberibacter asiaticus (CLas) acquisition assay over multiple Asian citrus psyllid (ACP) generations. CLas-free ACP adults were transferred to CLas -infected citrus plants (**A**). After a 15-day oviposition period (OP), adults were removed and the progeny development was monitored (**B**). Progeny was kept on those plants for 20 days after the emergence of adults (post-emergence—PE) for a new oviposition cycle (**C**). After this period, the adults were removed and the progeny development was monitored and maintained for 20 days (PE) for oviposition of the next ACP generation (**D**). This cycle was repeated four times (four ACP generations). ACP adults removed from CLas-infected plants in each generation were stored at −80 °C for CLas detection by qPCR.

**Figure 2 insects-11-00469-f002:**
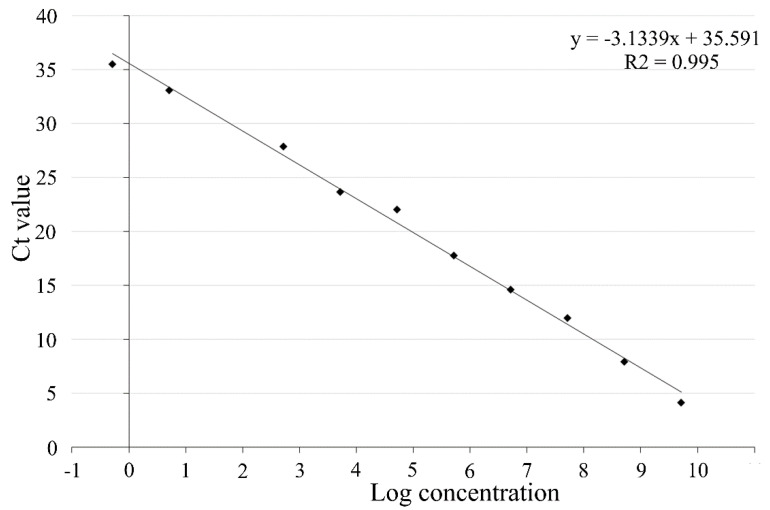
Molecular standard curve generated by real-time PCR for RpoB primer set through SYBR-green qPCR. Target DNA ranged from 10^9^ to 10^−1^ copies.

**Table 1 insects-11-00469-t001:** Estimation of the proportion of CLas positive ACPs (*p*) and the number of CLas cells per 100 ng of DNA on each generation analyzed. The *p* parameter was calculated by the formula *p* = 1 − I^(1/k)^, where I is the frequency of non-infected insects and k is the number of insects used per sample (Swallow,1985).

Generations	Total Number of Samples	Number of ACP per Sample	Number of Positive Samples (Frequency)	*p* (×100) (%)	Number of CLas Cells per 100 ng of DNA
Parental	15	5	6 (0.4)	9.71	4.79 × 10^0^d
F1	33	5	25 (0.75)	24.21	1.40 × 10^1^c
F2	38	5	33 (0.86)	32.51	6.85 × 10^1^b
F4	15	5	15 (1.0)	-	5.39 × 10^3^a

Different letters correspond to statistical difference in Ct among ACP generations by one-way ANOVA combined with Tukey’s test (*p* ≤ 0.01).

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
