# Peer review of "Development on Infected Citrus over Generations Increases Vector Infection by ‘Candidatus Liberibacter Asiaticus in Diaphorina citri"

_insects, 2020, doi:10.3390/insects11080469_

Round 1

Reviewer 1 Report

This paper reports results of an experiment showing that rates of infection and bacterial titer of CLas increases to 100% in Asian citrus psyllids after they have been reared on infected citrus trees for four generations. The authors attribute this increase to some combination of transovarial transmission, the circulative-propagative transmission mode, and short-term adaptation of the pathogen and vector possibly involving immune suppression and endosymbiosis. In general, the experimental design seems appropriate and well described and the interpretation of results seems reasonable. This study will hopefully inspire future studies that investigate the molecular mechanisms that lead to the steady increase in infection and bacterial titer from generation to generation in the vector. Overall, the English is acceptable but could be improved with some editing. For example, in the Figure 1 caption, the word “accompanied” should probably be replaced with “monitored”.  Paragraphs 5 and 6 on p. 6 also contain some awkward sentences.

It would be interesting to know whether there is any evidence from prior studies that CLas infection has any positive fitness effects on ACP or whether CLas-infected plants are more attractive to ACP and/or cause them to feed for longer periods than non-infected plants (or plants with lower CLas titer). Infection-induced changes in feeding behavior could help explain the increase in infection over generations.

Reviewer 2 Report

Manuscript insects-842305 entitled ‘Development on Infected Citrus over Generations Increases Vector Infection by ‘Candidatus Liberibacter asiaticus in Diaphorina citri’’ by Inaiara de Souza Pacheco and co-workers describes the interesting phenomenon of the increase of ‘Candidatus Liberibacter asiaticus’ in the psyllid vector Diaphorina citri, upon its rearing for several generation on CLas infected Citrus plants. Prevalence of infected insects emerging from eggs laid by the previous generation are reported, together with quantification of CLas loads in groups of adults of each generation up to the 4th one. The manuscript also describes the development of quantitative real time PCR protocol for the bsolute quantification of this pathogen in the psyllid.

The manuscript is generally well written, although in some cases English must be revised (see line by line list of modifications below). The experiment is well designed and organized, the results are well described, and the conclusions are supported by the obtained results. There are few issues in the Materials and methods section that may require some attention:

  1. The newly developed real time PCR protocol for quantification of the bacterium efficiently detects up to 10 target copies, according to the standard quantification curve, and this concentration corresponds to about Ct 35.. Authors say that all insects are infected starting from the F1 generation, but they also state that individuals with Ct>34 were considered as not infected. This probably happens only for the F0 generation (Materials and Methods, 2.3). This is a problem: there should be healthy insects with no Ct at all. PCR was run with 40 cycles, and lowest standard concentration had a Ct of about 36 (from plot in Fig 2. Samples with Ct>34 are probably infected, but with pathogen load below the quantification threshold of the system. Of course, to say this, healthy insects should not show any Ct, as well as no DNA PCR control (there is no statement on having used these controls. In this case, as Syber Green chemistry is used, a melting curve analyses would have been a good option to prove the specificity of amplicons obtained at CT34 or higher. Was this done at all?
  2. Also, Author compared Ct of samples without extrapolating the effective bacterial concentration (despite the amplification of a concentration curve). I think this option is correct, although quite unusual, as it does not provide immediate access to the real bacterial load in the samples. This is evident in the Result and Discussion section, where the CLas load in F2 psyllids is presented as ‘two times’ (twice) higher than that of F1 generation. F2 Ct is about 29, F1 CT is about 31. It is hard to decipher if this difference corresponds to double concentration of the target, This can be more easily done by comparing bacterial loads derived from the interpolation of the standard curve, rather than raw Ct. Also, expressing load differences in terms of CT differences does not provide clues on how the results of this work relate to CLas quantificaiton performed under other circumstances by other authors.
  3. In the Results and Discussion section, Authors state that transovarial CLas transmission has been reported and can explain ‘also contribute to increment, even in low rates, in bacterial levels across ACP generations’. I agree that transovarial transmission may be an important issue here, as this could explain the increase in CLas loads generation by generation. I do not understand why Authors seem to doubt about this. In the absence of transovarial transmission, or even in the case of transovarial transmission, I would think that continuous rearing on infected plants has an impact on the relationship between the insect and the bacterium, through i) increase of bacterium load due to massive transovarial transmission, so each new generation starts with a higher bacterial load as nymphs, and / or ii) continuous exposure to CLas affects the immune system of the adult, allowing it to support a much higher multiplication of the bacterium. Did the Author observe any fitness alteration between generations (e.g. did each generation laid the same number of eggs? Did the same number of nymph hatch at each generation? Did adult show different survival length F3 o F4 generations?). Authors should discuss this issue.

Minor issues: 

Everywhere in general in the text: I would advise to replace ‘titer’ with load.

Results/Discussion. This section could re-organized to avoid duplications (e.g. the less developed immune system of nymph is cited more than once)

‘particularly for acquisition by adults)[6],’: please add a space before [6]

‘adults of the parental generation showed a low pathogen titer (mean Ct = 33) (Fig. 3).’: Corresponding to how many cells? is this in line with previous reports?

‘observed an frequency of CLas-positive’: please change to ‘observed a frequency of CLas-positive’  ‘As far as I know the Swallow p value may help solving this issue

‘the 100% frequency observed in this work may be a result … on Citrus plants [34]. This observation is not really relevant to the topic and can be cancelled.

‘On the other hand, studies have been showed’: English revision needed.

‘On the other hand, studies have been showed’: English revision needed.

‘the frequency of qPCR-positive samples for this bacterium increase across the psyllid generations’: replace with ‘increased’

‘Figure 2. A molecular standard curve generated of Real-Time PCR for RpoB primer set’: Revise English of this caption

‘Both curves were performed using SYBR-green as described before’: replace title with ‘Absolute quantification of candidatus Liberibacter asiaticus’

‘2.3 Nuclei acid…’: pleace replace with ‘Nucleic acid…’

Caption to Figure 1: ‘the development of progeny was accompanied’; ‘the development of progeny was accompanied and maintained’: not clear what ‘accompany’ means here.

‘symptomatic sweet orange trees individualized inside screened cages’: please rephrase, as it is not clear what it means.

‘The ACP used in this experiment come from a CLas-free colony’: please replace with ‘The ACP used in this experiment came from a CLas-free colony’

 The progression of symptoms culminate in intense defoliation’: please replace with ‘  The progression of symptoms culminates in intense defoliation’

Author Response

1- Thank you for your comment. PCR protocol for quantification of the bacterium used in this work was previously validated using several CLas-positive and CLas-negative samples. Unfortunately, unspecific amplifications may occur in the end of qPCR reaction. Analyzing standard quantification curve we observed that Ct 35 extrapolate the limit of detection and the points that showed Ct>34 in this curve presented different melting temperatures compared to other points of the curve, indicating amplification of non-target product. So, we established Ct 34 as the threshold of our qPCR reaction. Positive and negative controls were used on each qPCR reaction.

2- Thank you for your comment. The Ct values were replaced by number of bacterial cells per 10 ng of DNA, as suggested.

3-Thank you for your comment. We agree that transovarial transmission may be important for the increment of bacterial load across ACP generations however, evaluation of ACP biology (e.g number of egg, number of nymph hatch, adult survival) was not evaluated in this work. 

Minor issues: 

Critique: Everywhere in general in the text: I would advise to replace ‘titer’ with load.

Response: Thank you for your comment. Done.

Critique: Results/Discussion. This section could re-organized to avoid duplications (e.g. the less developed immune system of nymph is cited more than once)

Response: Thank you for your comment. R&D was re-organized. 

Critique: particularly for acquisition by adults)[6],’: please add a space before [6]

Response: Thank you for your comment. Done. 

Critique:‘adults of the parental generation showed a low pathogen titer (mean Ct = 33) (Fig. 3).’: Corresponding to how many cells? is this in line with previous reports?

Response: Thank you for your comment. The Ct values were replaced by number of bacterial cells (per 100ng of DNA).

Critique: ‘observed an frequency of CLas-positive’: please change to ‘observed a frequency of CLas-positive’  ‘As far as I know the Swallow p value may help solving this issue

Response: Thank you for your comment. The Swallow analysis was added to R&D.

Critique: ‘the 100% frequency observed in this work may be a result … on Citrus plants [34]. This observation is not really relevant to the topic and can be cancelled.

Response: Thank you for your comment. Done. 

Critique: ‘On the other hand, studies have been showed’: English revision needed.

Response: Thank you for your comment. The manuscript was completely revised, and the English correction was performed.

Critique: ‘the frequency of qPCR-positive samples for this bacterium increase across the psyllid generations’: replace with ‘increased’.

Response: Thank you for your comment. Done. 

Critique: ‘Figure 2. A molecular standard curve generated of Real-Time PCR for RpoB primer set’: Revise English of this caption.

Response: Thank you for your comment. Done. 

Critique: ‘Both curves were performed using SYBR-green as described before’: replace title with ‘Absolute quantification of candidatus Liberibacter asiaticus’.

Response: Thank you for your comment. Done. 

Critique: ‘2.3 Nuclei acid…’: pleace replace with ‘Nucleic acid…’.

Response: Thank you for your comment. Done. 

Critique: Caption to Figure 1: ‘the development of progeny was accompanied’; ‘the development of progeny was accompanied and maintained’: not clear what ‘accompany’ means here.

Response: Thank you for your comment. Word ‘accompany’ was replaced by monitored.

Critique: ‘symptomatic sweet orange trees individualized inside screened cages’: please rephrase, as it is not clear what it means.

Response: Thank you for your comment. The phrase was re-organized.

Critique: ‘The ACP used in this experiment come from a CLas-free colony’: please replace with ‘The ACP used in this experiment came from a CLas-free colony’

Response: Thank you for your comment. Done. 

Critique: The progression of symptoms culminate in intense defoliation’: please replace with ‘  The progression of symptoms culminates in intense defoliation’.

Response: Thank you for your comment. Done. 

Reviewer 3 Report

The authors investigated a biological parameter of the pathosystem APC-CLas that could play important role in the spreading of the bacterium in citrus orchards, that is, the effect of continuous exposition for several generations of the vectors to infected plants on the infection rate and bacterial population of APC. The work is quite well structured, addressing one simple question and providing some data to answer it. Nonetheless, I found several inaccuracies and some major flaws, especially in statistical analysis, that could theorically influence the goodness of your results.

Dear authors,

I have attached the pdf of the MS with several little corrections, here I write some comments on the major flaws that need to be addressed in my opinion.

Comment # 1

Experimental setup described in chapter 2.2 is quite unclear on some key aspects. In sentence “Groups of 14 CLas-free ACP adults within 10 days of emergence were transferred to five CLas-infected and symptomatic sweet orange trees individualized inside screened cages”, do you mean that ACP adults were isolated in groups of 14 individuals per each single sweet orange tree? What do you mean with “individualized”, the five trees were separated from each others, each placed in a different cage? These are very important aspects that can influence the correct statistical analysis, and thus the results outcome. It is not very clear to me, please rephrase this paragraph to make your experiment more understandable to the reader.

Comment # 2

If I have well interpreted the setup, you have isolated groups of ACPs on different plants, then followed them across generations on the same plant. This seems to me a clear case of pseudoreplication, thus a mixed model (LMM or GLMM) would be more appropriate to conduct the statistical analysis. In case I misunderstood you explanation, and you had all the ACPs in the same cage, or being the trees true replicas, then a one-way ANOVA is fine to evaluate the relationship between bacterial titer and ACP generations, but please add a sentence that confirm you controlled for overdispersion and that there were no violations to the ANOVA assumptions.

Comment #3

Tukey test it is not ANOVA, but a post-hoc multiple comparison test used to compare the means of different groups, please describe it more accurately.

Comment #4

You did not perform any statistical analysis on the percentage of positive ACPs. There are several methods to analyse percentage data, I would suggest to use a binomial GLM using logit link on proportion data. Moreover, the same concerns on pseudoreplication I had exposed in Comment #2 apply here.

Comment #5

Why data on F3 is not shown in Fig. 3 or described in the text? Please add it.

Comment #6

You tested ACPs in pooled samples of five insects each, but I miss how you linked the PCR results on pooled samples to the experimental population. That is, if the pooled sample resulted positive to PCR how did you consider it? I could not find an explanation of your method neither in M&M or results. If you had considered all the five insect as positives, or considering a pooled group as single observation, this is wrong, since you should take into account the presence of some positive individuals within a positive pooled sample. A method to deal with this uncertainty is p = 1- H^(1/k) (p of Swallow, see for example (https://www.apsnet.org/publications/phytopathology/backissues/Documents/1987Articles/Phyto77n10_1376.PDF). In case you did not consider it, please recalculate the proportion of positives in your experiment.

Comment #7

I am not sure Fig. 2 is necessary, usually it is reported only the equation of the standard curve that is sufficient to describe your results. Also, since you obtained a standard curve, I suggest to express the results (CLas titer) as copies/100 ng DNA. Presenting data in this way is a more correct way to describe the bacterial load present in the vectors; furthermore this number is directly proportional to the increase of bacterial load, while Ct values are not, thus are more difficult to be immediately understood by the reader.

Author Response

Comment # 1: Experimental setup described in chapter 2.2 is quite unclear on some key aspects. In sentence “Groups of 14 CLas-free ACP adults within 10 days of emergence were transferred to five CLas-infected and symptomatic sweet orange trees individualized inside screened cages”, do you mean that ACP adults were isolated in groups of 14 individuals per each single sweet orange tree? What do you mean with “individualized”, the five trees were separated from each others, each placed in a different cage? These are very important aspects that can influence the correct statistical analysis, and thus the results outcome. It is not very clear to me, please rephrase this paragraph to make your experiment more understandable to the reader.

Response: Thank you for your comment. The sentence was rephrased. The five trees were separated from each other, each placed in a different cage.

Comment # 2: If I have well interpreted the setup, you have isolated groups of ACPs on different plants, then followed them across generations on the same plant. This seems to me a clear case of pseudoreplication, thus a mixed model (LMM or GLMM) would be more appropriate to conduct the statistical analysis. In case I misunderstood you explanation, and you had all the ACPs in the same cage, or being the trees true replicas, then a one-way ANOVA is fine to evaluate the relationship between bacterial titer and ACP generations, but please add a sentence that confirm you controlled for overdispersion and that there were no violations to the ANOVA assumptions.

Response: Thank you for your comment. The five plants are true replicas so, we used one-way ANOVA. The data do not violate the ANOVA assumptions Normality test was performed to confirm the normal distribution of the data.  This information was added on M&M (Lines 131/132).

Comment #3: Tukey test it is not ANOVA, but a post-hoc multiple comparison test used to compare the means of different groups, please describe it more accurately.

Response: Thank you for your comment. Both ANOVA and Tukey test were performed. This information was rephrased in the text (Lines 130/131; 157/158).

Comment #4 and #6

You did not perform any statistical analysis on the percentage of positive ACPs. There are several methods to analyse percentage data, I would suggest to use a binomial GLM using logit link on proportion data. Moreover, the same concerns on pseudoreplication I had exposed in Comment #2 apply here.

You tested ACPs in pooled samples of five insects each, but I miss how you linked the PCR results on pooled samples to the experimental population. That is, if the pooled sample resulted positive to PCR how did you consider it? I could not find an explanation of your method neither in M&M or results. If you had considered all the five insect as positives, or considering a pooled group as single observation, this is wrong, since you should take into account the presence of some positive individuals within a positive pooled sample. A method to deal with this uncertainty is p = 1- H^(1/k) (p of Swallow, see for example (https://www.apsnet.org/publications/phytopathology/backissues/Documents/1987Articles/Phyto77n10_1376.PDF). In case you did not consider it, please recalculate the proportion of positives in your experiment.

Response: Thank you for your comment. The swallow analysis was performed and the proportion of insect positives was added in the text (lines 128/129; 152-160; 191-195).

Comment #5: Why data on F3 is not shown in Fig. 3 or described in the text? Please add it.

Response: Thank you for your comment. The biological material of F3 was lost in a lab incident, the number of remained samples was insufficient to perform the analysis. 

Comment #7: I am not sure Fig. 2 is necessary, usually it is reported only the equation of the standard curve that is sufficient to describe your results. Also, since you obtained a standard curve, I suggest to express the results (CLas titer) as copies/100 ng DNA. Presenting data in this way is a more correct way to describe the bacterial load present in the vectors; furthermore this number is directly proportional to the increase of bacterial load, while Ct values are not, thus are more difficult to be immediately understood by the reader.

Response: Thank you for your comment. The Ct values were replaced by number of bacterial copies/100 ng DNA, as suggested. 

Reviewer 4 Report

The manuscript “Development on Infected Citrus over Generations Increases Vector Infection by ‘Candidatus Liberibacter asiaticus in Diaphorina citri’” by De Souza Pacheco et al. shows that the frequency of Candidatus Liberibacter-positive adult samples increases over generations when psyllids are constantly reared on CLas-infected citrus.

The work, although not of outstanding novelty, is well written and is a nice piece of work. The experimental plan is scientifically sound and conclusions are supported by the results. However, in my opinion, the way the quantitative results are expressed can be improved, as the authors presented them in a basic way.

I have two major suggestions for the improvement of the manuscript.

  • why not transforming "number of positive samples" into estimated proportion of positive insects according to Swallow? (Swallow WH 1985 Group testing for estimating infection rates and probabilities of disease transmission. Phytopathology 75:882-889). This would allow to better estimate the actual proportion of insects that acquired the bacterium. I suggest to replace number of positive samples with estimated proportion of infected insects overall the manuscript, text and figures
  • I strongly suggest to provide the estimation of bacterial load in the insects using a measure different from Ct cycle. Using Ct (threshold cycle in qPCR) is a very rough and basic way of estimating target quantity in quantitative PCR. One cannot understand why the authors included in their analysis (correctly!) a standard curve made of dilutions of the plasmid with the cloned fragment, and then expressed the bacterial load using Ct! A very simple calculation taking into account the number of copies of the RpoB gene in the CLas genome (possibly one, I do not know, I am not an expert of Liberibacter) should allow to estimate the number of bacteria cells in the samples, otherwise, what would be the purpose of the standard curve? Also, personally I do not like to establish thresholds based on Ct to discriminate positive and negative samples (e.g. “Samples of Ct > 34 were considered CLas-negative”. This is a bit trivial, although we have to read it in several papers….. The authors used a Sybr-Green based assays, so that they can check melting curve to assess specificity of the amplification signal, or at least define the threshold based on a given number of cells estimated using the standard curve. The general concept is that, when including a standard curve in each qPCR assay, each plate can be analyzed independently and it makes no sense to establish a “universal” threshold for all the plates, as different PCR assays runs a bit differently).

Minor concerns are:

Replace “replication” (of the bacterium) with “multiplication”. Replication is for viruses

I suggest to replace “bacterial titer” with bacterial load or amount. Titer is not fully correct according to some vector entomologists (including myself)

Add also in mat&met the information about the number of insect groups analyzed by qPCR (15 to 34). It is only stated in caption to figure 3, it should be included in mat&met in the paragraph 2.3

Figure 1 is useful, but the back arrow in the bottom linking panel D with panel C was misleading to me. Try to find another option to represent the repetition of this last phase

It is not clear to me what is the meaning of “development of progeny was accompanied”. Maybe “accompany” is not the proper verb for that

In the Results and Discussion section only the immune system is evoked to explain different acquisition ability of nymphs and adults. I suggest to consider also the different behavior/feeding behavior. Adults are much more mobile and likely to feed for shorter periods, while nymphs spend most of the time feeding. If the authors agree with my view (I am not an expert of Diaphorina and I can be wrong), I suggest to include this possible explanation, besides the immune-based one. In my experience, starting from a less efficient acquisition (e.g. shorter feeding time on an infected plant), the pathogen load in the insect vector remains lower for a long time, in spite of its multiplication in the insect body.

Author Response

Critique: why not transforming "number of positive samples" into estimated proportion of positive insects according to Swallow? (Swallow WH 1985 Group testing for estimating infection rates and probabilities of disease transmission. Phytopathology 75:882-889). This would allow to better estimate the actual proportion of insects that acquired the bacterium. I suggest to replace number of positive samples with estimated proportion of infected insects overall the manuscript, text and figures. 

 Response: Thank you for your comment. As suggested, Swallow analysis was performed and the proportion of positive insecs was added in the text (lines: 128-129; 152-158; 191-194). 

Critique: I strongly suggest to provide the estimation of bacterial load in the insects using a measure different from Ct cycle. Using Ct (threshold cycle in qPCR) is a very rough and basic way of estimating target quantity in quantitative PCR. One cannot understand why the authors included in their analysis (correctly!) a standard curve made of dilutions of the plasmid with the cloned fragment, and then expressed the bacterial load using Ct! A very simple calculation taking into account the number of copies of the RpoB gene in the CLas genome (possibly one, I do not know, I am not an expert of Liberibacter) should allow to estimate the number of bacteria cells in the samples, otherwise, what would be the purpose of the standard curve? 

Response: Thank you for your comment. The Ct values were replaced by number of bacterial cell/100 ng DNA, as suggested. 

Critique: Also, personally I do not like to establish thresholds based on Ct to discriminate positive and negative samples (e.g. “Samples of Ct > 34 were considered CLas-negative”. This is a bit trivial, although we have to read it in several papers….. The authors used a Sybr-Green based assays, so that they can check melting curve to assess specificity of the amplification signal, or at least define the threshold based on a given number of cells estimated using the standard curve. The general concept is that, when including a standard curve in each qPCR assay, each plate can be analyzed independently and it makes no sense to establish a “universal” threshold for all the plates, as different PCR assays runs a bit differently).

Response: Thank you for your comment. Both parameters cited by you (melting curve and number of cells) were used to estimate the Ct threshold. Analyzing standard quantification curve we observed that Ct 35 extrapolate the limit of detection and the points that showed Ct>34 in this curve presented different melting temperatures compared to other points of the curve, indicating amplification of non-target product. So, we established Ct 34 as the threshold of our qPCR reaction.

Minor concerns:

Critique: Replace “replication” (of the bacterium) with “multiplication”. Replication is for viruses. 

Response: Thank you for your comment. Done.

Critique: I suggest to replace “bacterial titer” with bacterial load or amount. Titer is not fully correct according to some vector entomologists (including myself). 

Response: Thank you for your comment. Done.

Critique: Add also in mat&met the information about the number of insect groups analyzed by qPCR (15 to 34). It is only stated in caption to figure 3, it should be included in mat&met in the paragraph 2.3.

Response: Thank you for your comment. The number of insect groups analyzed was added on Table 1 (lines: 152-158). 

Critique: Figure 1 is useful, but the back arrow in the bottom linking panel D with panel C was misleading to me. Try to find another option to represent the repetition of this last phase.

Response: Thank you for your comment. Figure 1 was remade.

Critique: It is not clear to me what is the meaning of “development of progeny was accompanied”. Maybe “accompany” is not the proper verb for that. 

Response: Thank you for your comment. The sentence was rephrased and the English were revised. 

Critique: In the Results and Discussion section only the immune system is evoked to explain different acquisition ability of nymphs and adults. I suggest to consider also the different behavior/feeding behavior. Adults are much more mobile and likely to feed for shorter periods, while nymphs spend most of the time feeding. If the authors agree with my view (I am not an expert of Diaphorina and I can be wrong), I suggest to include this possible explanation, besides the immune-based one. In my experience, starting from a less efficient acquisition (e.g. shorter feeding time on an infected plant), the pathogen load in the insect vector remains lower for a long time, in spite of its multiplication in the insect body.

Response: Thank you for your comment. R&D was re-organized and feeding behavior was added (lines 185-187). Indeed, nymphs spend most of the time feeding compared to adults, and this is mean reasons that nymphs acquire CLas more efficiently than Adults.  However, CLas persistence withing ACP seems to be related to immune system. Bacterial load tend to decrease when acquisition occurs on adults, while bacteria load increases when acquisition occurs on nymphs.